# PREFERENCE LEARNING FROM PHYSICS-BASED FEEDBACK: TUNING LANGUAGE MODELS TO DESIGN BCC/B2 SUPERALLOYS

## ABSTRACT

We apply preference learning to the task of language model generation of novel structural alloys. Where prior work focuses on generating stable inorganic crystals, our approach optimizes for the synthesizeability of a specific structural class: BCC/B2 superalloys, an underexplored family of materials with applications in extreme environments. Using three open-weight models (LLaMA-3.1, Gemma-2, and OLMo-2), we demonstrate that language models can be optimized for multiple design objectives using a single, unified reward signal through Direct Preference Optimization (DPO). Our reward signal is derived from thermodynamic phase calculations, offering a scientifically-grounded feedback for model tuning. To our knowledge, this is the first demonstration of preference-tuning a language model using physics-grounded feedback for targeted properties (in our case, BCC/B2 alloys). The resulting framework is general and adaptable to any design problem for which the design space is enumerable and simulation-based feedback is available.

## 1 INTRODUCTION

Materials discovery is challenging because of large design spaces sparsely covered by empirical results, and the intrinsic nonlinearity and multiobjectivity of materials design problems. Computational materials science addresses this sparsity by modeling from simulations, often based on density functional theory (DFT) (Kohn et al., 1996), and knowledge bases such as the Inorganic Crystal Structure Database (ICSD) (Zagorac et al., 2019). When trained on these sources, discriminative machine learning models can cheaply predict properties of unknown materials (forward design), while generative models can propose materials with favorable properties (inverse design).

Large language models (LMs), when trained or prompted appropriately, can generate descriptions of new materials. They are held as a potential accelerant to material discovery for their ability to draw on parametrically-encoded and retrieved domain knowledge to propose materials more likely to have desirable properties (Li et al., 2025; Brodnik et al., 2023). Prior work on LM-driven inverse design mostly falls into two categories. The first trains smaller local LMs, mostly via supervised fine-tuning (SFT) to generate candidate materials satisfying a single basic criterion, commonly thermodynamic stability (Gruver et al., 2024; Sriram et al., 2024; Antunes et al., 2024). The second category involves using a larger API-based LM as part of a search/optimization procedure to identify high-quality outputs according to multi-objective criteria, often in an multi-agent setup (e.g. Gan et al. (2025); Yang et al. (2024); Lai & Pu (2025)).

In this paper, we explore an intermediate step: using preference tuning to align local language models toward more optimal arbitrary downstream property values. Specifically, we use offline preference learning based on multiobjective feedback from a physical simulation model to nudge the LM into a "high-reward" output space where its generations are more likely to be of high quality while still remaining diverse within the chosen design space.

We apply this approach to the task of structural alloy design, specifically BCC/B2 "superalloys" consisting of a matrix of disordered, body-centered cubic (BCC) material surrounding precipitates of ordered BCC (B2) material. This type of alloy, consisting of two distinct phases, is a promising recent direction in extreme-environment structural alloys. By adding a second phase, they potentially address the structural weakness that existing alloys tend to exhibit at high temperatures (>1000°C)

(Kube et al., 2024b; Wang et al., 2018; Yurchenko et al., 2021). However, inducing the stable formation of two complementary phases is nontrivial. Any generative modeling approach needs to produce candidates that are both practically viable as well as potentially useful. Our approach generates superalloy candidates in the form of a composition for the BCC matrix, the B2 precipitate, and a suggested volume percentage for the B2. We apply a two-step modeling process mirroring conventional LM preference alignment. Starting with a known set of BCC and B2 compositions, we apply supervised fine-tuning (SFT) to three local instruction tuned language models (LLaMA 3.1 8B, Gemma-2-9B OLMo-2-7B) to produce (BCC/B2/B2 volume %) triples. We then use feedback on generated candidates from Thermo-Calc (Andersson et al., 2002), a popular thermodynamic simulation tool, to produce a multiobjective reward score for each candidate based on expert-designed heuristics. Finally, we use these scores for direct preference optimization (DPO), to push the models into a higher-reward output mode.

In our evaluation, we demonstrate that our SFT-tuned models are capable of generating valid alloy compositions that uniformly span the design space and exhibit novelty with respect to both the training data and existing entries in the Materials Project database. We further show that the DPO-tuned models, with the exception of OLMo, demonstrate improved average reward scores while retaining a high degree of diversity in their outputs. Our findings indicate that local language models can be effectively optimized for multiple design objectives using a single, unified reward signal. By comparison, larger state-of-the-art API-based LMs are able to suggest high-reward alloy compositions without tuning, but tend to hyper-fixate on specific elements and combinations, leading to limited exploration of the specified design space, a behavior resistant to prompt engineering. We conclude by outlining key takeaways and discussing how this preference tuning framework can potentially be extended to future materials discovery tasks and other domains within the physical sciences.

In summary, our contributions are as follows:

1. To our knowledge, this work presents the first instance of preference tuning for language models to generate materials compositions aligned with a practical, multiobjective design goal beyond basic thermodynamic stability.
2. We propose a general and extensible framework for scientist-informed candidate generation in non-parametric design spaces, leveraging offline feedback from physics-based simulations.[1]
3. We apply our framework to a real-world challenge in materials design—specifically, the discovery of BCC/B2 superalloys, moving away from general-purpose stable crystal generation toward targeted, high-impact alloy design.

## 2 RELATED WORK

**Conventional superalloy discovery**   Superalloys are a class of multiphase alloys that combine a ductile matrix phase with high-strength precipitates to produce a material that is both strong and tough at elevated temperatures. Current commercial superalloys, such as the Inconel and René classes of alloys, have a face-centered-cubic (FCC) matrix and $L1_2$ intermetallic precipitates. However, modern operation demands have now extended to temperatures beyond the design limit of any known FCC/$L1_2$ superalloy. In the search for even higher temperature alloys, significant interest has been directed at systems composed of a body-centered-cubic (BCC) matrix with ordered B2 precipitates, due to their prevalence in high-temperature refractory and multi-principal element alloys (Begley et al., 1968; Hobson, 1962; Naka & Khan, 1997; Wang et al., 2018). However, while some progress has been made in targeted studies (Frey et al., 2022; 2024; Kube et al., 2024a; Li et al., 2020; Ma et al., 2017; Shaysultanov et al., 2017; Wang et al., 2022; Whitfield et al., 2020), the enormity of the design space for BCC/B2 alloys strongly motivates the use of artificial intelligence for discovery.

Historically, the development process for new alloys has been slow, often requiring more than a decade, due to complex iterative experimental loops. Recent advances in *ab-initio* simulations, such as density functional theory (DFT) (Kohn et al., 1996) and molecular dynamics (Bartolotti & Flurchick, 1996; Geerlings et al., 2003; Humphrey et al., 1996; Kresse & Furthmüller, 1996a;b; Kresse & Hafner, 1993), have accelerated the materials discovery and enabled extensive ground-truth databases of stable compounds (typically containing 3 or more elements) (Curtarolo et al., 2012; Jain et al., 2013; Saal et al., 2013). However, the properties of such multi-element alloys depend on beyond-atomistic

---

[1]Code and data available at [redacted for anonymity]

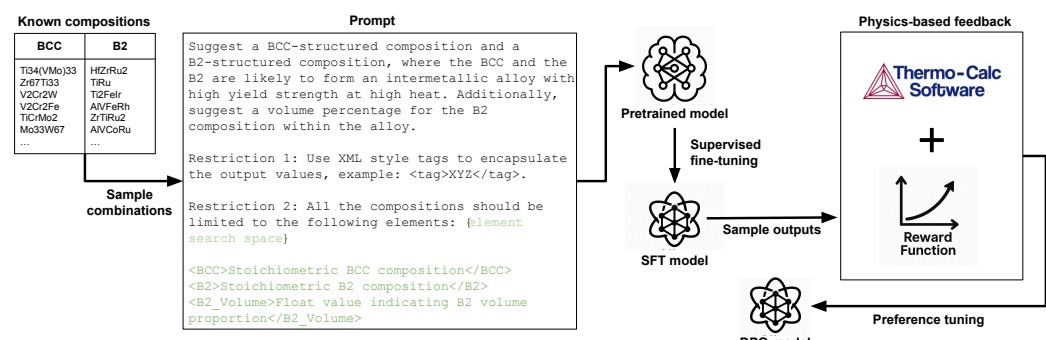

Figure 1: Schematic representation of training the language model for alloy design starting from a pre-trained language model using SFT, physics-based feedback, and DPO.

level dynamics. Computational alloy discovery relies more on thermodynamic simulation methods such as CALPHAD (CALculation of PHAse Diagrams). CALPHAD uses bulk-scale calculations of competing free energy curves to determine the material phases that will be stable at a given temperature and composition. CALPHAD has been applied to alloy development as early as the 1970s (Kaufman & Nesor, 1974), and modern software packages such as Thermo-Calc (Andersson et al., 2002) make high-throughput calculations for alloy screening relatively straightforward. Simulations like DFT and CALPHAD are commonly used as feedback for algorithmic optimization loops such as Bayesian Optimization (Vela et al., 2023; Hastings et al., 2025).

**Language models for materials**  Most recent AI-driven materials discovery efforts use graph neural networks (GNNs), which excel as discriminative predictors from structured representations (forward design). Merchant et al. (2023) exemplifies the forward design approach, employing a greedy algorithm to generate candidate compounds, which are then evaluated for thermodynamic stability using a GNN. Several other studies explore the application of GNNs to predict material properties (Chen et al., 2023; 2019). By contrast, inverse design begins with a target set of properties and aims to generate novel material candidates expected to exhibit those properties. Gruver et al. (2024) demonstrates that local LMs can be fine-tuned from a dataset of stable crystals to produce novel stable crystals, similarly to (Antunes et al., 2024). This focus on thermodynamic stability has characterized most other recent work this area (Sriram et al., 2024; Yang et al., 2024; Yan et al., 2024). Perhaps the most similar recent paper to the present effort, PLaID (Xu et al., 2025b), applies DPO to Llama-7b to improve stability of generated crystals. More recent work Cao & Wang (2025); Xu et al. (2025a), focuses on adding constraints like novelty and uniqueness in the preference tuning objective other than stability. However, as Seshadri & Cheetham (2024) note, generating thousands of stable materials is not practically useful for working materials scientists. The importance of utility is missing from present work. Another limitation of many of previous studies is the use the crystallographic information file (CIF) format, which has both intrinsic downsides (Xiao et al., 2023) and little direct representation in e.g., scientific literature, raising questions about what useful biases LMs can bring to CIF-based inverse design tasks. Other recent work leverages LMs without doing any parametric optimization, often via agentic approaches (Lai & Pu, 2025; Gan et al., 2025; Yang et al., 2024).

## 3 METHOD

Prior work in LM-generated materials has focused primarily on producing stable inorganic crystals (e.g. Gruver et al. (2024); Xu et al. (2025b)), evaluating which requires the model to produce Crystallographic Information File (CIF) files specifying the structure of the candidate crystal. In this work we focus on a narrower design space (combination of known BCC/B2 pairs) where stability is implied by construction and our physical feedback mechanism requires only the composition as input. Therefore, we train our models to generate compositions rather than full CIF descriptions.

Our approach involves three key steps. First, we construct a cold-start dataset of (BCC/B2/B2 volume %) triples and use it to train a model via supervised fine-tuning (SFT), enabling it to explore the

full alloy design space. Next, we sample candidates from the SFT model and evaluate them using Thermo-Calc for thermodynamic feedback. Finally, we use this feedback to define a hierarchical reward function based on ability to laboratory-synthesize, and apply Direct Preference Optimization (DPO) to align the model with expert-guided preferences. Figure 1 illustrates this pipeline.

**Supervised fine-tuning (SFT)**   To build our dataset for SFT, a list of 207 known BCC and 88 known B2 compositions (both binary and ternary) was collected from the Materials Project (CC-BY 4.0)(Jain et al., 2013) and filtered based on stability and alloying suitability (Andersson et al., 2002; Thermo-Calc Software, (Accessed May 2025). These elements and compounds were then combined in various proportions to form hypothetical alloys and verified for thermodynamic feasibility using the Thermo-Calc (SUNLL) (Andersson et al., 2002) TCHEA7 database (DSUNLL) (Thermo-Calc Software, (Accessed May 2025) to produce ground-truth triplets of the form (BCC/B2/B2 volume %) . The SFT dataset consists of all possible pairwise combinations of these compositions (18,216 distinct pairs), combined with three B2 volume percentages for each, sampled from a normal distribution with a mean of .45, capped at .20 and .70, for a total size of 54,648 examples. Additional details can be found in Appendix A.3.

We tune the SFT model using a causal language modeling (CLM) objective, using an instruction-based prompt (Figure 1). To reduce the number of trainable parameters, we employ low-rank adapter modules (LoRA) Hu et al. (2022), configuring the adapters with a rank of 8 and scaling factor $\alpha = 32$. This setup results in only  0.027% (for LLaMA) and  0.057% (for OLMo) of parameters being updated during fine-tuning. Following  Gruver et al. (2024), we introduced special tokens to the tokenizer vocabulary (if they did not exist) for padding, beginning of sentence, end of sentence, and unknown to properly tokenize chemical formulas. More details about the training can be found in Appendix A.2. The generations sampled from this stage are combined into a master composition based on the molar volume percentage of B2 and fed to Thermo-Calc to assess thermodynamics.

**Reward function**   Preference feedback for DPO comes from the Thermo-Calc tool (Andersson et al., 2002), which takes as input a single composition and temperature and, using a combination of simulation and databases of empirical results, predicts what phases are likely to exist in what quantity at that temperature. To create a reward score for an SFT-generated (BCC/B2/B2 volume %) triple, we use the B2 volume % to combine the BCC and B2 compositions into a single master composition, then query Thermo-Calc on this composition at a range of temperatures from $373K$ to $2273K$. An example of output from Thermo-Calc is shown in Appendix Figure 9.

Realizing a fabricable superalloy requires multiple interplaying factors to align during processing, namely: (i) the BCC phase must be the first to solidify from a liquid melt; (ii) the B2 phase should form at a temperature below that of the BCC phase, but still at as high of a temperature as possible to maximize the thermal operation limit of the alloy; (iii) the alloy must be comprised entirely (or nearly entirely) of BCC and B2, as other intermetallic compounds are often brittle and weak, making them largely undesirable; and (iv) the BCC and B2 phases should have nearly identical crystal lattice dimensions, which reduces the build-up of internal stresses in the alloy during processing and use. We operationalize these viability rules as follows (in descending order of importance):

1. There must be some temperature at which both a solid BCC and B2 phase exist simultaneously. (`bcc_b2_exist`)
2. The BCC must form first as the temperature decreases. (`bcc_forms_first`)
3. A B2 phase must exist close to room temperature, $373K$. (`b2_room_temp`)
4. No more than 10% of non BCC/B2 phases should form at any temperature. (`others_exceed_10%`)

When all these criteria are satisfied, the quality of a candidate is measured as the minimum difference in lattice parameter (reported in Å) between BCC and B2 phases at any temperature (`min_lattice_mismatch`). This mismatch value typically varies from $10^{-1}$ to $10^{-7}$. The overall reward is numericized as a weighted sum of indicators for these boolean conditions:

$$\text{Reward}(\text{BCC}, \text{B2}, \text{Volume}) =$$
$$- 1000\,\mathbf{1}_{\neg\texttt{bcc\_b2\_exist}} - 100\,\mathbf{1}_{\neg\texttt{bcc\_forms\_first}}$$
$$- 10\,\mathbf{1}_{\neg\texttt{b2\_room\_temp}} - \mathbf{1}_{\texttt{others\_exceed\_10\%}}$$
$$- \texttt{min\_lattice\_mismatch} \tag{1}$$

The reward score ends up negative log-scaled, with a worst possible score of $\sim -10^3$ and best of $\sim -10^{-7}$, with $> -10^0$ being the viability threshold of obeying the four basic rules. These coefficients reflect a tiered prioritization of synthesis realism: thermodynamic coexistence is fundamental, while lattice mismatch offers fine-grained selection. Ultimately the score reflects the **viability and potential for favorable properties of the candidate BCC/B2 alloy**, rather than a direct estimate of its properties per se. This reflects the way CALPHAD calculations are used in traditional alloy design (e.g. Holgate et al. (2025)), as a screening filter on potential candidates. Since this class of materials is in its infancy, the reward function does not target high-temperature performance, instead focusing on candidates with favorable properties at any temperature in range. It could easily be made more specific by, for instance, setting a minimum temperature threshold on the various rules to ensure that they hold at the target conditions.

**Direct preference optimization (DPO)**   To guide our model toward producing higher-quality (BCC/B2/B2 volume %) triples, we sample candidates $S_{\theta_{SFT}}$ from the SFT model and calculate their reward score using Eq. 1. From the output of our reward function we create a pairwise preference dataset $\mathcal{D}_{\text{DPO}}(y^+, y^-)$, where $y \in S_{\theta_{SFT}}$ indicating a preferred generation $(y^+)$ over $(y^-)$. We want to push our model towards a region of higher rewards by optimizing a contrastive objective, reviewed more fully in the appendix, where hyperparameter $\beta$ controls the distance between the distribution of the original SFT model distribution and that of the the new model. We want the internal reward mapping of the model (as no separate reward model is required in DPO) to learn from our multiobjective reward scores and push the model to search the parametric space of higher average reward. However, to prevent the preference tuned model from going wildly out of distribution or hacking the reward function (Rafailov et al., 2024), we set $\beta = 0.5$. Training was conducted using a low-rank adapter module, trained for 1 epoch (more details in A.3).

For the DPO dataset, we sample 5,000 (BCC/B2/B2 volume %) triples from the SFT model, then use Thermo-Calc to compute a scalar reward for each generation. We construct a preference dataset with the top 25% generations, as ranked by reward, paired with 100 randomly selected lower ranked generations. This strategy allows the model to learn from relative preferences, encouraging discrimination between high- and low-quality outputs.

## 4 EXPERIMENT

**SFT and DPO models**   We perform SFT and DPO on three open instruction-tuned LMs of comparable size: LLaMA-3.1-8B (Grattafiori et al., 2024), Gemma-2 (9B) (Team et al., 2024), and OLMo-2-7B (OLMo et al., 2024). We use low-rank adapters ($\alpha = 32, rank = 8$) for training, with 8-bit quantized models.

**Baselines**   To properly evaluate the gains and limitations of our approach, we compare it against several varying strong baselines. **(1) Random search**: Our first baseline mimics traditional parametric search by randomly sampling a subset of compositions from a grid of BCC and B2-forming elements, with the B2 molar volume sampled uniformly between 20% and 70% (more details in Appendix A.1). **(2) Prompting API-based models**: We use few-shot prompting of state-of-the-art (at the time of writing) API-based large LMs, including GPT-4.1, GPT-O3, and Gemini-2.5. Prompts are available in the Appendix. **(3) Prompt tuning**: We find empirically (see below) that prompting approaches suffer from poor diversity in their outputs. To create a stronger baseline, we extend the most balanced API model (Gemini-2.5) and automatically tune the input prompt to encourage diversity, using the MIPROv2 optimization method from the DSPy library (Khattab et al., 2023). **(4) Agentic approach**: To further investigate the capabilities of the API based models we create a simple agentic system where two agents: a generator and an evaluator work in conjunction to come up with high quality alloy compositions. The generator agent generates a composition and the evaluator accepts or rejects the composition with a feedback. We optimize the generator via verbal reinforcement from the evaluator agent. More details in Appendix. **(5) Prior published models**: Additionally, we incorporate generations from previously published generative models, including Crystal-LLM (Gruver et al., 2024) and CDVAE (Xie et al., 2021), which aim to generate crystal structures of inorganic compounds. Although these models are trained for general-purpose stable inorganic crystals, we filter their outputs to retain only those compositions that fall within our target alloy design space, i.e., potential BCC/B2 alloy composed of TCHEA elements.

| Model | Validity | Coverage Recall | Coverage Precision | Novelty | Mean Reward | Unique pairs @100 |
|---|---|---|---|---|---|---|
| Random search | 0.80 | 0.98 | 0.82 | 0.44 | -883.71 | 1.0 |
| CDVAE | 0.73 | 0.43 | 0.07 | 0.94 | – | – |
| Crystal-LLM-7B | 0.90 | 0.34 | 0.18 | 0.80 | – | – |
| Crystal-LLM-13B | 0.87 | 0.44 | 0.17 | 0.81 | – | – |
| Crystal-LLM-70B | 0.91 | 0.45 | 0.17 | 0.83 | – | – |
| GPT-4.1 | 1.00 | 0.32 | 1.00 | 0.86 | -53.23 | 0.44 |
| GPT-O3 | 1.00 | 0.42 | 1.00 | 0.99 | -75.43 | 0.66 |
| Gemini-2.5 | 0.99 | 0.79 | 0.99 | 0.81 | -106.22 | 0.82 |
| Prompt-tuned Gemini-2.5 | 0.99 | 0.83 | 1.00 | 0.98 | -350.34 | 0.91 |
| Agentic GPT-4.1 | 1.00 | 0.48 | 1.00 | 0.77 | -542.60 | 0.98 |
| Agentic Gemini-2.5 | 1.00 | 0.78 | 1.00 | 0.87 | -19.87 | 0.61 |
| Gemma SFT | 0.99 | 0.99 | 1.00 | 0.94 | -220.41 | 0.98 |
| Llama SFT | 0.99 | 0.99 | 0.99 | 0.92 | -215.92 | 0.99 |
| OLMo SFT | 0.99 | 0.99 | 0.99 | 0.92 | -218.54 | 1.00 |
| Gemma DPO | 1.00 | 0.95 | 1.00 | 0.97 | -206.71 | 0.92 |
| Llama DPO | 0.99 | 0.98 | 1.00 | 0.93 | -175.89 | 1.00 |
| OLMo DPO | 0.99 | 0.98 | 1.00 | 0.95 | -268.72 | 0.98 |

Table 1: Evaluation of generative models on validity, coverage, and novelty as proposed by Xie et al. (2021), as well as mean reward score and what fraction of 100 generated BCC/B2 pairs are unique (lower indicates more self-repetition).

## 5 EVALUATION

### 5.1 BASIC RESULTS

Our basic results, shown in Table 1, use compositional validity, coverage, and novelty metrics, as introduced by Xie et al. (2021) and later adopted by Gruver et al. (2024). Compositional validity is assessed using the Pauling electronegativity test, which ensures that the constituent elements exhibit appropriate electronegativity differences (Davies et al., 2016). Coverage is computed as the Euclidean distance between the normalized feature vectors of generated compositions and all 18,216 potential BCC/B2 alloy compositions–coverage recall measuring what percentage of the space is produced, and coverage precision measuring what percentage of produced compositions belong within the space. Novelty is measured as the pairwise distance between generated samples and all known (existing) alloys containing two or more TCHEA elements, based on their feature representations. While coverage measures how well the generated compositions span the known design space, novelty captures how different they are from all existing alloys. We also report mean reward score among generated compositions, and "Unique pairs @100", the fraction of 100 generated BCC/B2 pairs that are unique. A lower score on this latter value indicates more self-repetition and less diversity. Following prior work, we use Matminer (Ward et al., 2018) to vectorize the compositions. We sample at least 1000 generations from each model with $\tau = 1.0$. **An ideal model should have near-perfect validity and achieve a balance between coverage, novelty and reward.**

From Table 1, we observe that general-purpose crystal generation models struggle to produce valid BCC/B2 alloys within our defined design space. These models show low coverage recall and precision, frequently missing key regions of the space and generating chemically irrelevant compositions, over half of which fail the compositional validity checks. Randomly sampling from existing BCC and B2 compositions leads to a high coverage but the final result is often (about 30% times) not a valid composition and not a BCC/B2 alloy for about 20% of the time. Novelty also goes down since they are similar to existing alloys in the MP database.

Among the API-based models, the generated compositions demonstrate high validity and coverage precision, often near perfect. However, they exhibit low coverage recall and low pair uniqueness, meaning that they tend to repeat themselves while failing to fully span the design space. Their relatively high novelty scores indicate they produce compositions distinct from those in the Materials Project database. They produce high-reward candidates, especially GPT-4.1, indicating that their retrieved/parametric knowledge provides useful biases, though these biases presumably also prevent them from exploring certain regions of the design space, hence the lower coverage.

We find it difficult to improve diversity in API model output without sacrificing mean reward. The prompt-tuned Gemini-2.5 model, whose prompt is optimized toward generating diverse outputs, demonstrates higher coverage and pair uniqueness than the other API-based models, but this comes at the cost of reward, with its proposed alloys underperforming even the SFT models. Between the two agentic baselines, GPT-4.1 shows a similar improvement in diversity balanced against a collapse in mean reward, while agentic Gemini-2.5 demonstrates the best mean reward of any approach we tried, at the expense of output diversity.

The local SFT models, trained on a uniform sample of (BCC/B2/B2 volume %) triples, are all comparable. They demonstrate high validity, coverage, novelty and pair uniqueness. This indicates that they succeed at becoming a "blank slate", generating uniformly from the designated space of possible (BCC/B2/B2 volume %) triples. While this doesn't make them very useful alloy-proposers on their own, it does make them suitable for further optimization toward a specific goal, which we implement in the form of DPO.

## 5.2 EFFECT OF PREFERENCE TUNING

Table 1 shows that the DPO models, with the exception of OLMo, show a modest improvement in mean reward over their SFT precursors, while maintaining their high coverage of the design space and generated pair uniqueness. Their mean reward is lower than that of the API based models (excluding prompt-tuned Gemini-2.5), indicating that they learn fewer biases than these larger models.

Figure 2 illustrates the effect of DPO with Win/Draw/Loss analysis based on reward score. Gemma and LLaMA DPO models win 49.8% and 52.1% of the time and lose 46.1% and 45.4% of the time, respectively. The rest were draws. However, the OLMo DPO model lost to its SFT counterpart 52.4% of the time and won only 42.3% of the time.

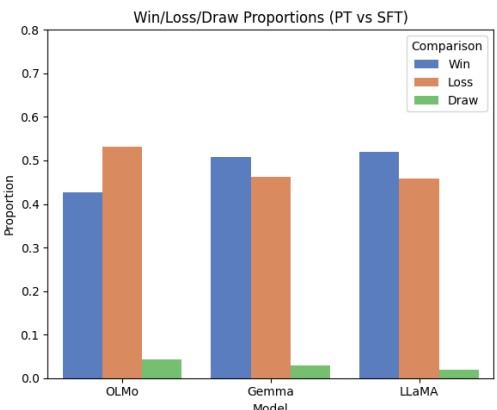

Figure 2: Each bar represents the proportion of cases where the DPO model outperformed (Win), underperformed (Loss), or matched (Draw) its SFT counterpart in reward score.

Figure 3: Percentage change in objective satisfaction from SFT to DPO models across Gemma, OLMo, and LLaMA. The plot illustrates the relative improvement or degradation in meeting four alloy design objectives after preference tuning (DPO).

Figure 3 assesses how effectively the cumulative learning signal optimized the models for individual synthesis objectives. We evaluate the four manually-chosen subcomponents of the reward function: (1) BCC and B2 phases must coexist at some temperature; (2) BCC must form first at a higher temperature; (3) B2 must exist at room temperature; and (4) BCC/B2 phases must be present across 90% of the evaluated temperature range. We compute the percentage change in the satisfaction rate—defined as the proportion of generated alloys that satisfy each objective—from the SFT to the DPO models. As shown, all synthesis objectives improve in LLaMA, while three out of four improve in Gemma. In contrast, OLMo exhibits degradation across all four objectives following preference tuning. Two key insights emerge from these results: (1) optimizing for the presence of the B2 phase at room temperature remains challenging, as both Gemma and OLMo perform worse on this criterion,

and LLaMA shows only modest improvement; and (2) combining multiple reward signals in this setup can push certain architectures like OLMo off-distribution, leading to a collapse in performance across objectives, possibly due to its smaller capacity or mismatch with reward distribution.

## 5.3 Hyperfixation in API-based models

API-based models such as GPT-4.1 and Gemini-2.5 models are powerful and easy to use, which begs the question of whether local models have a place in LM-driven materials discovery alongside API-based models and the agentic systems built on top of them. Our analysis in Section 5.1 shows that the strong biases of these models limits their coverage recall and generated BCC/B2 pair uniqueness. To better understand this limitation, we conduct a focused analysis to understand patterns of hyperfixation in their behavior.

| Rank | GPT-4.1 | | GPT-O3 | | Gemini-2.5 | | Prompt-tuned Gemini-2.5 | | Llama DPO | | Llama SFT | |
|---|---|---|---|---|---|---|---|---|---|---|---|---|
| | Elements | Freq | Elements | Freq | Elements | Freq | Elements | Freq | Elements | Freq | Elements | Freq |
| 1 | {Mo, Nb} | 0.500 | {Mo, Nb, W} | 0.578 | {Mo, Nb} | 0.145 | {Mo, Nb, Ta} | 0.115 | {Mo, Nb, Ti} | 0.072 | {Cr, Ti, V} | 0.041 |
| 2 | {Nb, W} | 0.382 | {Mo, Nb, Ta, W} | 0.152 | {Mo, Nb, W} | 0.136 | {Mo, Nb, Ti} | 0.096 | {Mo, Nb, W} | 0.048 | {Ti, V, W} | 0.038 |
| 3 | {Mo, Nb, W} | 0.105 | {Mo, Ta, W} | 0.140 | {Nb, W} | 0.089 | {Mo, Nb, Ta, Ti} | 0.059 | {Nb, Ti, W} | 0.048 | {Nb, Ti, V} | 0.037 |
| 4 | {Cr, Mo, W} | 0.008 | {Mo, Nb, V, W} | 0.045 | {Nb, Ta, W} | 0.073 | {Mo, Nb, Ta, W} | 0.054 | {Mo, Ti, W} | 0.046 | {Mo, Ti, V} | 0.036 |
| 5 | {Mo, Nb, Ta} | 0.001 | {Mo, Nb, Ta} | 0.020 | {Cr, Mo, W} | 0.062 | {Mo, Ta, W} | 0.052 | {Cr, Mo} | 0.040 | {Mo, Nb, W} | 0.033 |

Table 2: Top 5 most frequent BCC element combinations generated by each model.

Table 2 explains the prompting model result by showing the top 5 BCC element combinations generated by a selection of models. We can see that half of few-shot GPT-4.1's BCCs are Mo/Nb combinations, and 98% use some subset of Mo/Nb/W. Few-shot Gemini shows a similar but less extreme level of fixation, with at least 36% of its BCC candidates a subset of the same Mo/Nb/W combination. A prompt-tuned Gemini-2.5 few-shot approach reduced this even more, with about 13% BCC with some combination of Mo/Nb/Ta. By contrast, DPO LLaMA shows a much more even spread, only slightly more concentrated than SFT LLaMA. This means that the API models achieve high average reward by fixating on a small selection of elements and element combinations.

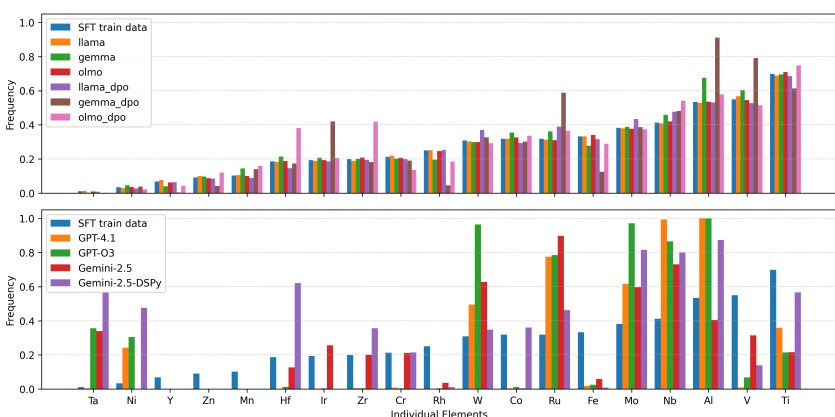

Figure 4: Output frequencies of individual elements by trained models (top) and API models (bottom), respectively, compared to the training data.

Finally, Figure 4 shows the distribution of individual elements favored by the SFT and DPO models versus the API models. The top plot shows that SFT and DPO generations have an element distribution similar to the training data. Among all the trained models we can see that DPO Gemma and DPO OLMo are fixating slightly more on some elements like Ir/Ru/Al/V and Hf/Zr/Nb/Ti, respectively. In particular DPO OLMo generated Hf and Zr at much higher frequency and Ir by DPO Gemma than the training compositions. The bottom plot shows the fixation of few-shot GPT-4.1 (green), Gemini-2.5 (red) and prompt-tuned Gemini-2.5 (violet) on certain elements like Ta/Ni/Hf/Zr/W while completely missing on elements like Y/Zn/Mn. Gemini is noticeably more adherent to the training data element frequencies than GPT-4.1, with GPT-4.1 hyperfixating on Nb and Al beyond what is in the training data.

The sum total of these results shows that API-based models achieve high reward by focusing on known high-reward regions, to the exclusion of unknown regions, and that this behavior is difficult to dislodge via prompt tuning or agentic iteration without badly affecting reward. It is widely acknowledged that pre-existing biases affect and limit exploratory materials development (Jia et al., 2019; Horgan, 2021), and our analysis seems to indicate that API-based models reflect those same biases. Therefore, there may be a role for models capable of learning useful reward signals while still retaining a high degree of exploratory openness, as our DPO-tuned models demonstrate.

# 6 DISCUSSION

Preference tuning is valued for its ability to optimize language models toward objectives that are (1) noisy and (2) hard to describe or articulate (such as politeness or humor). That makes it appropriate for optimizing LM-generate materials toward arbitrary physical objectives.

Our results show that DPO is able to produce a modest improvement in average reward while maintaining high diversity in output, for two of three local LMs. OLMo, on the other hand, performed worse after DPO across all objectives. We observe increased divergence of key token logits between SFT and DPO for OLMo, which explains the collapse (more analysis in Appendix A.6). While we apply our SFT-DPO training process to a highly specific design space and reward function, it is a highly general protocol, and could be applied to any engineering problem capable of using an SFT training set to represent a design space and with a computationally-efficient verifier available over generated candidates. One possible example is battery design, where open-source tools like PyBaMM (Sulzer et al., 2021) could be used to assess generated candidates.

While model training can identify good regions of feature space, black box optimization (BBO) is more suited to identifying standout candidates within that space. BBO methods such as Bayesian Optimization are a major part of computational alloy discovery (Hastings et al., 2025; Wang & Dowling, 2022), and recent work has sought to combine LMs with Bayesian Optimization as both generators of candidate points and discriminators over generated candidates (Liu et al., 2024; Chang et al., 2025). While the useful biases of API-based models makes them more likely to suggest high-reward candidates (when used as generators) and more likely to correctly assess provided candidates (when used as discriminators), their tendency to fixate on certain regions of feature space limits their ability to perform the "explore" part of the exploration/exploitation tradeoff in discrete optimization. Tuned local models offer a potential solution to this problem by offering more control over their degree of bias, particularly via the $\beta$ parameter of the DPO process.

**Limitations** One limitation of this work is that the predictions produced by Thermo-Calc and similar tools are not perfect, and become less reliable for many-element compositions in regions for which the tool's databases have poor coverage. Engineering a confidence estimate for external feedback, combined with LM reasoning over external context like prior scientific findings, could be a way of mitigating this issue, as could, in a fully realized modeling pipeline, the inclusion of physical experimentation to verify the predicted properties of key candidates. A higher-level limitation is the question of whether, for downstream DO tasks, a higher-reward baseline distribution is actually needed and worth the investment in time and effort to create. If our ultimate goal is to find a small number of exceptional alloy candidates, it might be more efficient to simply perform a search through the output space of the SFT model. Future work will explore this question.

**Conclusion** We apply preference tuning for the first time to LM-driven inverse design of materials toward functional properties, and propose preference-tuned "high-reward" models as an intermediate step toward LM-driven materials discovery. Our supervised fine-tuning is successful, while our preference tuning results are positive, though inconsistent between models. While we apply these ideas specifically to BCC/B2 superalloy discovery, the template we introduce here is general, and could be adapted to any design problem where it is possible to collect medium-scale feedback on model-suggested compositions, such as battery or photovoltaic materials Finally, this work is complimentary with other approaches for LM-guided materials discovery, such as agentic approaches, and could be extended to work as an improved baseline distribution for such methods.

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

# A   APPENDIX

We add more technical detail and approach of our work in here.

## A.1   BASELINES

Baseline approaches include (1) random search, (2) static prompting of API-based models, (3) automatic prompt tuning, and (4) a basic agentic setup.

### A.1.1   RANDOM SEARCH

Conventional alloy discovery approaches often do parametric sweeps of composition space for promising candidates. We approximate this approach by constructing a grid of BCC- and B2-forming elements and sampling random compositions from it. The BCC and B2 compositions were constructed separately by randomly sampling from the grid of constituent elements, and volume percentages individually with heuristic rules enforcing likely BCC- and B2-formation, e.g. that B2s must be a 1-to-1 ratio of two B2-forming elements. This method better imitates a traditional parametric search in conventional alloy discovery than randomly sampling known BCC/B2 pairs, as is done in the preparation of the SFT training data.

### A.1.2   PROMPTING

Our second baseline consists of one-shot and few-shot prompting of three state-of-the-art proprietary API-based models: Gemini-2.5, GPT-4.1 and GPT-o3. We find one-shot prompting from these models to be both dominated by few-shot prompting and unreliable in producing valid output formatting, so we do not report results from the former. In the zero-shot setting, we randomly sample a single exemplar from the SFT model output. In the few-shot setting, we provide top 10 and bottom 10 generations from the SFT model as exemplars, ranked on reward.

The prompts that we use for one-shot and few-shot prompting are provided in Figure 5 and Figure 6, respectively. The zero-shot prompting did not work because the models were unable to generate any feasible BCC-B2 pairs in a parseable format.

### A.1.3   PROMPT TUNING

Few-shot Gemini-2.5 produced the most favorable balance of diversity and reward amongst the prompting baselines. To see if this tradeoff could be further optimized, we create a prompt-tuned few-shot baseline using DSPy(Khattab et al., 2023), optimizing the prompt to produce diverse outputs. We used the MIPROv2 with "medium"-level optimization, using total number of TCHEA unique elements in the alloy system as the metric to optimize. During inference even when using a temperature of 1.0 we could not generate unique triplets with the tuned prompt, which shows robustness of the approach and good for a lot of things but not for us. To sample different composition triplets we added a Universally Unique Identifier (UUID) at the end of each prompt. As reported in table 1, this approach does improve diversity at the cost of mean reward.

### A.1.4   AGENTIC SETUP

We implement a simple agentic baseline consisting of a generator and evaluator, implemented with LangGraph (https://github.com/langchain-ai/langgraph). This is a simple setup where the generator agent is instructed to generate a (BCC/B2/B2 volume %) with few-shot prompt. The generated triple is then sent to the evaluator agent which grades it as "Valid" or "Invalid" generation, and also provides a detailed reason when judging invalid. If the generation is invalid then we re-route the feedback from the evaluator agent and ask the generator to re-generate the composition. We keep optimizing the generator when it produces invalid (BCC/B2/B2 volume %) for a maximum of five iterations. If the fifth generation is also invalid according to the evaluator we scrap the generation and restart the loop. Otherwise we add it to our acceptable alloy list (as in Figure 7). We run this generation-evaluation agentic loop independently until we get 1000 successful generations.

```
A BCC-B2 intermetallic alloy consists of a disordered
body-centered cubic (BCC) parent matrix and an ordered B2
precipitate, each existing in the material as some
fractional percentage. Suggest a BCC-structured
composition and a B2-structured composition, where the BCC
and the B2 are likely to form an intermetallic alloy with
high yield strength at high heat. Additionally, suggest a
volume percentage for the B2 composition within the alloy.

Restriction 1: Use XML style tags to encapsulate the
output values, example: <tag>XYZ</tag>.

Restriction 2: All the compositions should be limited to
the following elements: {element search space}

Example generation:
<BCC>Ti2Nb2Mo</BCC>
<B2>AlVFeCo</BCC>
<B2_Volume>51.45</B2_Volume>

Known BCC: {list of known BCC}
Known B2: {list of known B2}
```

Figure 5: This is the one-shot prompt we used for our API based models. We added some additional context while keeping the training prompt similar. The example generation was randomly sampled from our training data. The text in blue is optional.

This setup is commonly known as Evaluator-Optimizer [2]. Commonly this setup is used to produce high quality output, as it optimizes the output through iterative refinement [3]. In our use-case we wanted the model to produce high quality alloys and hence this setup made most sense (disregarding cost). A similar setup is used by Shi et al. (2025) to automate generation and refinement of simulation code for materials synthesis.

## A.2 SFT TRAINING DATA CURATION

To build our initial dataset of 207 body-centered cubic (BCC) and 88 B2-structured compositions, a list of known known BCC and B2 structures from the Materials Project (Jain et al., 2013), was filtered to keep only compounds comprised of the 26 elements in Thermo-Calc's TCHEA7 database (Thermo-Calc Software, (Accessed May 2025). A second filter was then applied to keep only compounds with a calculated energy above the convex hull between 0 and 0.25 eV/atom. (A compound with an energy of 0 eV/atom is expected to be stable at 0 K; by 0.25 eV/atom, a compound is highly unlikely to be stable at 0 K but could become stabilized by entropy effects at elevated temperatures relevant to BCC/B2 alloys.) This processing yielded 24 BCCs (primarily single-element entries) and 57 B2s (exclusively two-element pairs).

These lists served as the basis for further iteration. First, the role of all elements was estimated. For example, it was noted that elements like Nb and Mo generally formed stable BCCs, whereas Ti and Zr had larger energies above the convex hull and only form BCC structures at elevated temperatures. Likewise, for the B2 compounds, it was noted that elements like Al and Hf generally occupied the A-site, whereas Fe and Ru generally occupied the B-site; some elements, like Mn or V, could occupy either site, whereas others (e.g., Nb or Ta) were found in higher energy (less stable) B2s. These trends were used to iterate BCC compositions with element concentrations of 20%, 25%, 33%, 40%, 50%, 67%, or 75%; B2 compositions were iterated with 1–2 elements per site (at 25% or

---

[2]https://www.anthropic.com/engineering/building-effective-agents
[3]https://github.com/OmarKhaled0K/Agents_and_workflows?tab=
readme-ov-file

```
A BCC-B2 intermetallic alloy consists of a disordered
body-centered cubic (BCC) parent matrix and an ordered B2
precipitate, each existing in the material as some
fractional percentage. Suggest a BCC-structured composition
and a B2-structured composition, where the BCC and the B2
are likely to form an intermetallic alloy with high yield
strength at high heat. Additionally, suggest a volume
percentage for the B2 composition within the alloy.

Restriction 1: Use XML style tags to encapsulate the output
values, example: <tag>XYZ</tag>.

Restriction 2: All the compositions should be limited to
the following elements: {element search space}

Examples of good generations:

<BCC>TiV</BCC>
<B2>NbRu</B2>
<B2_Volume>34.8</B2_Volume>

.
.
.

<BCC>Nb67Mo33</BCC>
<B2>ZrTiRu2</B2>
<B2_Volume>50.6</B2_Volume>

Examples of bad generations:

<BCC>Zr33Ti67</BCC>
<B2>VRu</B2>
<B2_Volume>56.7</B2_Volume>

.
.
.

<BCC>Ti33Mo67</BCC>
<B2>AlVFe</B2>
<B2_Volume>64.75</B2_Volume>

Known BCC: {list of known BCC}
Known B2: {list of known B2}
```

Figure 6: This is the few-shot prompt we used for our API based models. We added some additional context while keeping the training prompt similar. Top-10 and bottom-10 of LLaMA SFT model generations were given here as examples of good and bad generations respectively (only two are shown here for brevity). The text in blue is optional.

50% concentration). A mixture of stable and metastable elements was used throughout this iteration process to ensure a broad representation of potentially stable phases. This process resulted in 2,413 potential BCC compositions and 1,101 potential B2 compositions. Each potential composition was

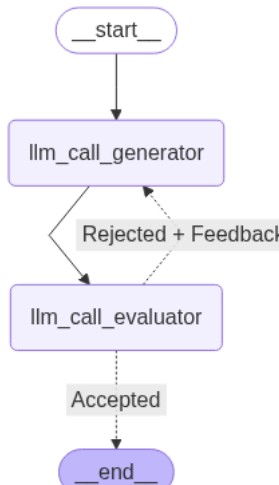

Figure 7: Simple agentic setup with LangChain. We keep the loop going for a maximum of five iterations.

evaluated with Thermo-Calc, and only compositions forming >99% BCC or B2 were kept, leaving 207 BCC and 88 B2-structured compositions used for SFT. Finally, a volume fraction of B2 intermetallic was prescribed by drawing from existing BCC-B2 alloys and domain expertise. We sampled the B2 volume percentage uniformly within the [20%, 70%] interval. Therefore, the supervised dataset consists of structured triplets of the form BCC, B2, B2 volume proportion. For each unique BCC–B2 pair, we sampled three distinct volume fractions, resulting in approximately 55,000 triplets. This dataset defines the compositional search space over which our language model operates.

## A.3 SFT TRAINING AND VALIDATION

Training was conducted with a batch size of 2 across three NVIDIA A40 GPUs with gradient accumulation every 4 steps. Finetuning was performed with 8-bit quantization and low-rank adapters ($rank = 8\ and\ \alpha = 32$) using the PEFT library [4]. The adapters were only added for "q_proj" and "v_proj", this yields maximum learning without parametric overhead (Hu et al., 2022). Cosine annealing was used as a learning rate scheduler. The entire training process required about 93 hours.

The training and evaluation performance for all three local models were similar, as show by loss curves ( Figure 8). Other than a higher starting point for OLMo, the loss curves are almost identical and converge quickly.

We trained each model for 5 epochs. While training loss plateaued after the first epoch, all three models showed steadily declining validation set loss until the end of training. The behavior of OLMo was more unstable than LLaMA or Gemma, but all models converged to a similar validation loss. The

## A.4 DPO TRAINING DATA CURATION

We sample 5000 (BCC/B2/B2 volume %) triples from each SFT model and evaluate them with Thermocalc. Thermocalc predicts the phases of an alloy master composition at different temperatures, resulting in a table where each row represents the predicted portion of a particular phase at a particular temperature (Figure 9).

We use this feedback to define a reward score for each composition (Eq. 1). The SFT-generated triples are then ranked in descending order by their reward. From this list, we select the top 25% (1250 triples), each designated as a chosen generation at index $i$. For every chosen generation, we randomly sample 100 distinct rejected generations from index positions with lower reward ($j > i$). This procedure yields $1250 \times 100 = 125,000$ preference pairs for DPO training.

---

[4]https://github.com/huggingface/peft

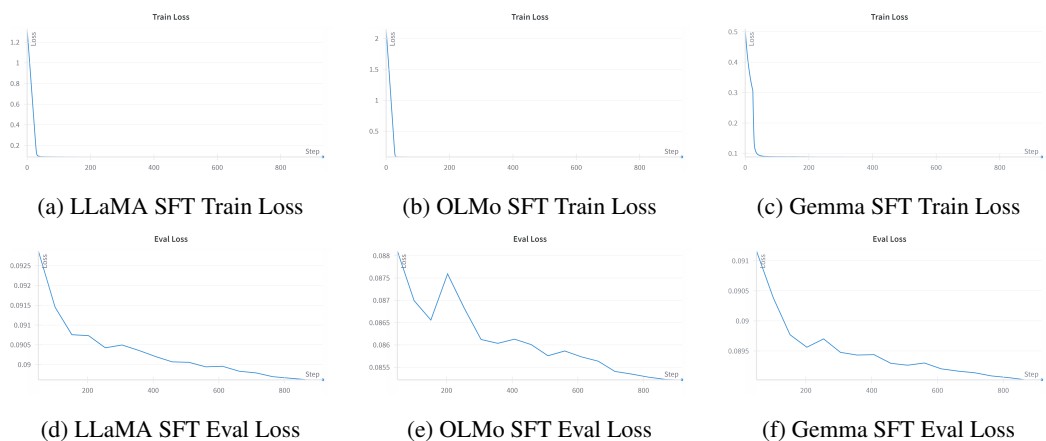

Figure 8: Loss curves for LLaMA, OLMo and Gemma during supervised fine-tuning (SFT).

We use this particular sampling strategy to balance quality against generalization. Variants we could have explored include a narrower definition of high-quality (e.g. top 10%), or pairing high-quality candidates against lower-quality candidates (e.g. worst 25%). We leave it to future work to optimize the sampling strategy for this type of approach.

| BCC | B2 | B2 Volume | Temperature | Quantity | Phase | IsOrdered | Lattice Parameter |
|------|------|------|------|------|------|------|------|
| Cr33Fe67 | MnAl2Fe | 61.65 | 373.15 | 0.16 | BCC_B2#1 | 0 | 2.91 |
| Cr33Fe67 | MnAl2Fe | 61.65 | 373.15 | 0.83 | BCC_B2#2 | 1 | 2.93 |
| Cr33Fe67 | MnAl2Fe | 61.65 | 1073.15 | 1 | BCC_B2#2 | 1 | 2.99 |
| Cr33Fe67 | MnAl2Fe | 61.65 | 1173.15 | 1 | BCC_B2#2 | 1 | 3.01 |
| … | … | … | … | … | … | … | … |
| Cr33Fe67 | MnAl2Fe | 61.65 | 2273.15 | 1 | LIQUID#1 | -- | -- |

Figure 9: Output from Thermo-Calc evaluates the stability of the generated BCC-B2 alloy over a range of temperatures. The reward function use this output to compute a scalar reward for preference tuning.

### A.5 DPO: TRAINING AND VALIDATION

For DPO, we take the adapter optimized with SFT and perform direct preference optimization. We train on the same configuration as SFT since this was our computational upper limit. We train each model for only 1 epoch. The DPO training took 70 hours to complete.

DPO optimizes the following objective:

$$\theta^* = \arg\min_{\theta} \sum_{(x,y^+,y^-)\in\mathcal{D}_{\text{DPO}}} \tag{2}$$

$$-\log \sigma\left(\beta \log \frac{\theta(y^+|x)}{\theta_{\text{SFT}}(y^+|x)} - \beta \log \frac{\theta(y^-|x)}{\theta_{\text{SFT}}(y^-|x)}\right)$$

$\theta_{\text{SFT}}$ and $\theta^*$ are model parameters of SFT and DPO models respectively, $\beta$ is the alternative to KL-penalty factor (Rafailov et al., 2023), which controls the distance between the distribution of the $\theta_{\text{SFT}}$ and $\theta^*$. We want the internal reward mapping of the model (as no separate reward model is required in DPO) to learn from our multiobjective reward scores and push the model to search the parametric space of higher average reward. However, to prevent the preference tuned model from going wildly out of distribution or hacking the reward function (Rafailov et al., 2024), we set $\beta = 0.5$.

The results from all models were again quite similar, with OLMo outperforming LLaMA in terms of reward margin on the evaluation set (Figure 10). We are unsure why OLMo failed to generate higher quality BCC/B2 compositions in spite of its better performance on the evaluation set.

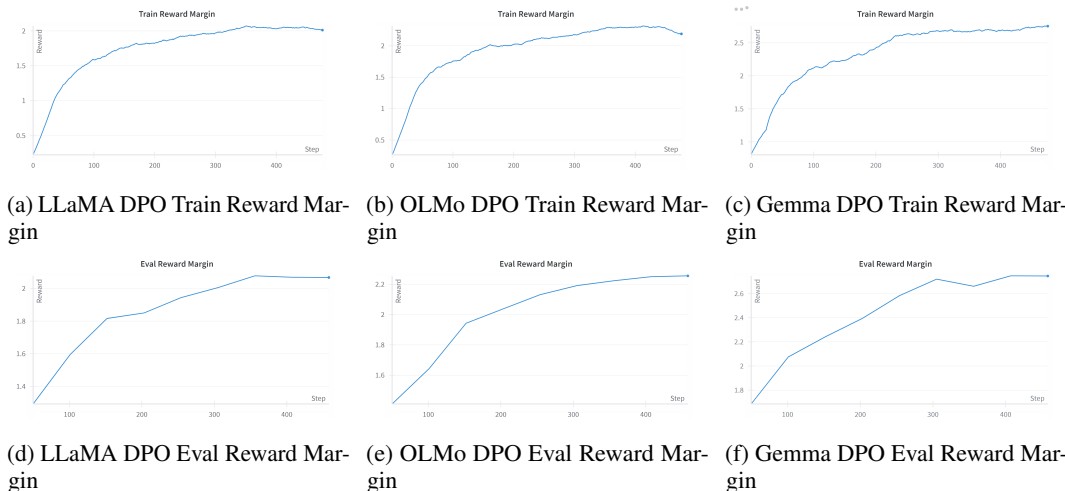

(a) LLaMA DPO Train Reward Margin

(b) OLMo DPO Train Reward Margin

(c) Gemma DPO Train Reward Margin

(d) LLaMA DPO Eval Reward Margin

(e) OLMo DPO Eval Reward Margin

(f) Gemma DPO Eval Reward Margin

Figure 10: Reward Margin for LLaMA, OLMo and Gemma during preference optimization (DPO).

### A.6 WHY PREFERENCE TUNING FAILED ON OLMO?

| Element | Count (OLMo) | KL (OLMo) | KL (LLaMA) | KL (Gemma) |
|---------|-------------|-----------|------------|------------|
| Ti | 94 | 0.0155 | 0.0078 | 4.25e–04 |
| Al | 53 | 0.0169 | 0.0104 | 3.34e–04 |
| V | 49 | 0.0122 | 0.0071 | 1.27e–04 |
| Nb | 38 | 0.0193 | 0.0030 | 3.33e–04 |
| W | 22 | 0.0015 | 0.00015 | 2.98e–05 |
| Cr | 13 | 0.0289 | 0.0257 | 1.32e–04 |

Table 3: Forward $D_{\mathrm{KL}}(\mathrm{DPO} \,\|\, \mathrm{SFT})$ on generated tokens (teacher–forced; trimmed at EOS) for the elements most frequently produced by OLMo. OLMo's KL is consistently higher than LLaMA's and far above Gemma's near-zero values, indicating model drift on domain-critical tokens.

**Why OLMo regressed while LLaMA and Gemma improved?** We diagnose the effect of preference tuning by measuring forward $D_{\mathrm{KL}}(\mathrm{DPO} \,\|\, \mathrm{SFT})$ strictly on the *generated continuation*: we teacher–force the SFT decode, trim at EOS, and compute KL token-wise. We also summarize KL over a *filtered token set* that carries the task semantics—element symbols and multi-digit numerals that encode compositions and phase fractions. Under this lens, **LLaMA** shows small, localized KL bumps at decision bottlenecks; **Gemma** remains close to its SFT policy; **OLMo** is different. Its KL spikes are both larger and more frequent, and they land exactly on the filtered tokens. In effect, the OLMo update reallocates probability mass on the symbols and numbers that define alloy identity, not on harmless stylistic tokens (see Table 3). This pattern naturally explains the downstream regressions: if the largest distributional shifts occur on element choices and volume proportions, the generator drifts off the "chemistry grammar" that SFT had learned, degrading satisfaction of the synthesis constraints.

**Interpretation from the KL profiles** The KL curves point to *over-steer* rather than lack of signal—a strength–sensitivity mismatch between the DPO update and OLMo's inductive bias. (1) *Architecture × adapter placement/rank:* the same LoRA targets and rank that are tame on LLaMA/Gemma appear to sit on more causal pathways in OLMo, so identical gradients yield larger effective steps in logits for rare technical tokens (elements, multi-digit numerals). (2) *Tokenizer/prior effects:* these tokens live in a low-frequency subspace; if OLMo's pretraining allocates less robust capacity there,

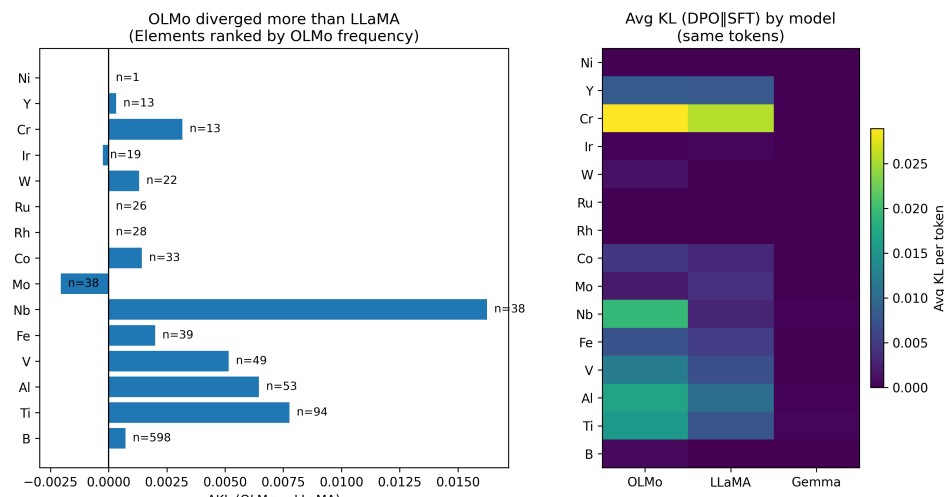

Figure 11: **OLMo goes out of distribution on domain-critical element tokens after DPO.** *Left:* Ranked bar chart of $\Delta\text{KL} = \overline{D_{\text{KL}}}(\text{DPO} \,\|\, \text{SFT})_{\text{OLMo}} - \overline{D_{\text{KL}}}(\text{DPO} \,\|\, \text{SFT})_{\text{LLaMA}}$ computed *only on generated tokens* (teacher-forced on the SFT continuation; trimmed at EOS). Elements are ordered by OLMo frequency; labels show OLMo occurrences ($n$). Positive bars indicate OLMo moved farther from its SFT reference than LLaMA did for the *same* token. *Right:* Heatmap of average per-token $D_{\text{KL}}(\text{DPO} \,\|\, \text{SFT})$ for the same elements across models (OLMo, LLaMA, Gemma). The consistently hotter OLMo column on key elements (e.g., Nb, Ti, Al, V) evidences over-steer in the chemistry subspace where alloy identity is decided, while LLaMA shows moderate shifts and Gemma remains near the SFT policy.

the preference gradients induce higher variance and numeric drift. (3) *DPO hyperparameters:* a $\beta$ and learning-rate/step schedule that gently nudges strong SFT policies (LLaMA/Gemma) can over-correct a weaker or more brittle SFT (OLMo), inflating KL precisely on the filtered token set. The net effect is the signature we observe: the biggest divergence occurs where correctness matters most (see Figure 11).

**Moving forward**    If we *weaken and stabilize* the update in that subspace—e.g., increase $\beta$ (gentler preference step), reduce LR/steps or LoRA rank, and/or retarget adapters (start with attention projections)—and optionally add a light reference anchor (DPO-KL or a small SFT CE mix-in), the filtered-token KL for OLMo should drop into the LLaMA band. Under the same teacher-forced evaluation, this KL reduction should coincide with recovery on the synthesis objectives. In short, the KL analysis localizes the failure mode (over-steer on domain-critical tokens) and directly suggests how to fix it.

