# OpenReview forum: "Preference Learning from Physics-Based Feedback: Tuning Language Models to Design BCC/B2 Superalloys"
_ICLR.cc/2026/Conference — ICLR 2026 Conference Withdrawn Submission_

### Official Review · Reviewer_CtH3 · 2025-10-25

**Soundness:** 2
**Presentation:** 3
**Contribution:** 1
**Rating:** 2
**Confidence:** 3

**Summary:**

This paper applies direct preference optimization (DPO) to fine-tune open-weight language models to generate BCC/B2 superalloy compositions, with Thermo-Calc simulations as feedback. The goal is to tune the generation of alloys toward satisfying criteria such as phase coexistence, solidification order, and lattice match. Experimental results show that DPO improves average reward and maintains compositional diversity, though inconsistent between LLaMA, Gemma, and OLMo; while untuned API-based models (GPT, Gemini) exhibit strong element fixation, limiting exploration.

**Strengths:**

DPO demonstrates a potentially useful approach to physics-based guidance to language models.

**Weaknesses:**

- The demonstrated use case, generating compositions and their volume fractions, is an oversimplified one in materials design, therefore not showing much value.
- The experiments and evaluations are not convincing (see Q1–2).
- The reward function relies heavily on heuristics, limiting general applicability (see Q3–4).

**Questions:**

1. A large part of “validity, coverage, and novelty” metrics mentioned in Sec. 5.1 is related to crystal structure, in addition to composition. Since the proposed method generates only composition, I don’t feel the metric is useful here.
2. The comparison with general-purpose crystal generators like Crystal-LLM and CDVAE don’t seem fair, as they are neither designed for nor fine-tuned on the specific task.
3. The reward function in Equation (1) is largely hand-crafted. Is the choice of scaling factors quantitatively rigorous, or just qualitatively? How to generalize the proposed method to other application scenarios?
4. Following up on Q3, if an accurate reward function is known, is LLM a good method for the materials design synthesis task? How would it compare to reinforcement learning (RL) or Bayesian optimization (BO)?

---

### Official Review · Reviewer_SWw1 · 2025-11-01

**Soundness:** 3
**Presentation:** 3
**Contribution:** 3
**Rating:** 4
**Confidence:** 2

**Summary:**

This paper introduces a framework for designing novel BCC/B2 superalloys by fine-tuning language models (LMs) using preference learning. The LMs are optimized via Direct Preference Optimization (DPO), uniquely employing a reward signal derived from multi-objective, physics-based feedback from thermodynamic simulations (Thermo-Calc). This work is the first to align LMs toward a practical engineering goal using physics-grounded feedback, moving beyond simple stability to optimize for complex, synthesizeable materials.

**Strengths:**

1. The paper is clearly written and well-motivated, convincingly arguing for a shift from optimizing simple stability to complex engineering utility.

2. It demonstrates that preference learning is a promising pathway to achieve this, successfully aligning language models with physics-grounded, multi-objective design goals.

**Weaknesses:**

1. The study's core contribution, preference learning via DPO, yielded only modest gains over the SFT baseline. This method proved inconsistent, as it failed on one of the three test models (OLMo), which showed significant performance degradation. This undermines the claim of a successfully applied and robust preference learning framework.

2. The paper only tested DPO and failed to explore advanced methods such as GRPO, which is widely used in training recent reasoning models. This is a significant omission, especially given the modest results of DPO.

**Questions:**

Please refer to the Weaknesses section above.

---

### Official Review · Reviewer_jX7K · 2025-11-01

**Soundness:** 2
**Presentation:** 3
**Contribution:** 2
**Rating:** 4
**Confidence:** 3

**Summary:**

The paper proposes a pipeline to use LLMs for designing two-phase BCC/B2 superalloys. The paper starts from a thermo-calc–generated dataset, combining 200+ BCC compositions and ~90 B2 compositions into ~50k synthetic training instances. They first supervised finetuned several open LLMs (LLaMA-3.1, Gemma-2, OLMo-2) to reliably produce compositions in this structured schema. Then they sample new candidates from the SFT models, evaluate them with Thermo-Calc over a temperature range, and build a hierarchical, physics-based reward that prioritizes: (1) two-phase BCC+B2 presence, (2) BCC solidifying first, (3) B2 still present at room temperature, and (4) wide two-phase stability window. These rewards are converted into pairwise preferences and used to run DPO-style preference tuning.

**Strengths:**

1. Instead of materials generation in general, the paper targets “one-shot generation of a BCC composition, a B2 composition, and a B2 volume fraction” for a concrete, high-value family of alloys (BCC matrix + ordered B2 precipitates), which makes the problem easy to evaluate.

2. Using physics-generated preferences (from Thermo-Calc) to do DPO on top of an SFT model is a reasonable and novel adaptation of recent preference-learning/LLM-alignment techniques to materials design.

3. The paper actually prompts large API models and even uses prompt-optimization (e.g., DSPy/MIPRO-like) to show that generic LLMs tend to “hyper-fixate” on a narrow element set, whereas the proposed pipeline preserves diversity.

4. The overall pipeline: synthetic SFT data from a simulator $\rightarrow$ sampling $\rightarrow$ physics-based reward $\rightarrow$ pairwise DPO could be reused for other material design problems that have an automated simulator.

**Weaknesses:**

1. The same thermo-calc setup is used to (1) synthesize SFT data, (2) generate preference pairs, and (3) evaluate success. This makes it hard to tell whether the model learned transferable “materials knowledge” or just learned to speak to this particular CALPHAD database. Cross-simulator or literature-based sanity checks are missing.

2.  Only some base models (e.g., LLaMA, Gemma) improve stably after physics-DPO, while others (e.g., OLMo-2) degrade, which suggests the pipeline is sensitive to model architecture or to off-distribution sampling. This weakens the “general framework” claim.

3. The paper mostly reuses validity/coverage/novelty-style metrics from crystal/material generation, but the target here includes a continuous volume fraction and a thermo-mechanical notion of “two-phase stability over a temperature sweep.” A metric that directly measures the percentage of candidates that satisfy all four thermo-based sub-objectives would be more convincing.


4. No experimental or literature recovery check. There is no demonstration that the model can rediscover known BCC/B2 alloy compositions from the literature, or that any predicted alloy is plausible under process constraints (cooling rate, aging). That makes the current contribution look more like a simulation-level template than an end-to-end materials design result.

**Questions:**

1. Are all preference pairs derived from the same Thermo-Calc database/settings that were used to generate the SFT data? If so, can you provide any cross-tool or cross-database validation to show the model is not overfitting one CALPHAD implementation?

2. Some base models (e.g., OLMo-2) do not benefit from physics-based DPO. Did you try alternative preference-learning setups ($\beta$-DPO, listwise preferences, GRPO/PPO with the same reward) to stabilize training across models?

3. In Table comparisons, API LLMs with good prompt engineering still look competitive. Do those API models produce samples that are judged better by Thermo-Calc because they encode extra “chemical intuition” not present in your synthetic SFT data? If yes, can high-reward API samples be distilled back into the local model?

4. Since you claim the framework is general, can you show at least one non-BCC/B2 or non-superalloy example (even a small one) to demonstrate that replacing Thermo-Calc with another physics/simulation engine would still work?

---

### Official Review · Reviewer_nCXy · 2025-11-01

**Soundness:** 3
**Presentation:** 3
**Contribution:** 1
**Rating:** 2
**Confidence:** 4

**Summary:**

This study proposes an approach to fine-tuning LLMs for materials design by incorporating feedback from physical simulations. First, information derived from existing materials databases was utilized to enhance the model's ability to predict material properties through supervised fine-tuning (SFT). Subsequently, performance indicators obtained from physical simulations were employed in direct preference optimization (DPO), enabling model training toward the generation of materials with desired properties. Through this two-stage learning strategy, the effectiveness of the method was validated in maintaining the chemical and physical validity of generated material candidates while optimizing their performance. As a result, the proposed framework introduces a new paradigm for materials design that integrates physical constraints and feedback into data-driven models, representing an important step toward autonomous material discovery through the synergy of simulation and artificial intelligence.

**Strengths:**

- A major strength of this study lies in its focus on the important challenge of new materials discovery using generative AI, representing a true integration of scientific research and AI technology. In particular, by applying language models to materials design, the study demonstrates the potential for knowledge-integrated materials exploration that goes beyond conventional data-driven approaches.

- The approach of fine-tuning large language models (LLMs) with feedback from physical simulations is highly interesting. It successfully balances validity with generative diversity, achieving reliable materials design through the synergy between AI and physical modeling.

- Another notable strength is the practical application of the proposed method to real-world materials discovery tasks, accompanied by detailed evaluations of generated results and their underlying factors.

**Weaknesses:**

- This study presents a highly application-oriented approach to materials design that combines physical simulations with large language models (LLMs), but it offers limited novelty in terms of machine learning methodology itself. Therefore, it may be more suitable for publication in a specialized journal in the fields of Materials Informatics or Computational Materials Science, rather than at a top-tier machine learning conference such as ICLR, which prioritizes methodological innovation.

- While the authors provide thorough analyses and discussions from the perspective of materials science, the study does not introduce new technical developments regarding model architecture or optimization methods. Although the integration of physical simulations is interesting, advances in learning algorithms and model design remains limited, resulting in a relatively modest contribution to AI research.

- Another limitation is that the weighting of the reward function and other optimization parameters appear to be determined through empirical trial and error rather than clear theoretical reasoning. Consequently, it is uncertain to what extent the reported results demonstrate the general validity of the proposed approach, and further investigation into reproducibility and applicability is needed.

**Questions:**

- Since this paper is submitted to a top-tier machine learning conference such as ICLR, a certain level of novelty and ingenuity in the machine learning methodology is expected. While the study is valuable as an applied research, it would be helpful to clarify how domain-specific knowledge and constraints from materials science perspective are incorporated into the algorithm design and training process.

- How were the hyperparameters — such as the weighting of the reward function, the dimensionality of LoRA, and the learning rate and batch size for SFT and DPO—optimized? Could you explain in detail whether these values were determined empirically through trial and error or based on theoretical or experimental approaches.

- It would also be beneficial to present concrete examples of materials generated by the proposed approach and discuss their scientific significance or value from a materials science perspective. If the generated structures or properties exhibit scientific validity or novelty, providing both quantitative and qualitative evaluations would strengthen the discussion.

---

### Note · Authors · 2025-11-20

**Comment:**

We thank the reviewers for their helpful feedback. Due to the scope of critiques and requested changes, we are choosing to withdraw this paper for consideration from ICLR 2026.

**Withdrawal Confirmation:**

I have read and agree with the venue's withdrawal policy on behalf of myself and my co-authors.